

# Coherence locking in a parallel NMR probe defends against gradient field spillover

Mengjia He[1], Neil MacKinnon[1*], Dominique Buyens[1], Burkhard Luy[2,3] and Jan G. Korvink[1*]

[1]*Institute of Microstructure Technology, Karlsruhe Institute of Technology, Eggenstein-Leopoldshafen, Germany.*

[2]*Institute for Biological Interfaces 4 - Magnetic Resonance, Karlsruhe Institute of Technology, Eggenstein-Leopoldshafen, Germany.*

[3]*Institute of Organic Chemistry, Karlsruhe Institute of Technology, Karlsruhe, Germany.*

*\* Corresponding authors: neil.mackinnon@kit.edu, jan.korvink@kit.edu*

## Abstract

The implementation of parallel nuclear magnetic resonance detection aims to enhance measurement throughput in support of high throughput screening applications including, for example, drug discovery. In support of modern pulse sequences and solvent suppression methods, it is important that each detection site has independent pulsed field gradient capabilities. Hereby, a challenge is introduced, in which the local gradients applied in parallel detectors introduce field spillover in adjacent channels, leading to spin dephasing and hence to signal suppression. This study proposes a compensation scheme employing optimized pulses to achieve coherence locking during gradient pulse periods. The design of coherence-locking pulses utilizes optimal control to address gradient-induced field inhomogeneity. These pulses are applied in a PGSE experiment, and a parallel HSQC experiment, demonstrating their effectiveness in protecting the desired coherences from gradient field spillover. This compensation scheme presents a valuable solution for magnetic resonance probes equipped with parallel and independently switchable gradient coils.

## 1 Introduction

While nuclear magnetic resonance (NMR) technology is routinely employed to analyze large sets of chemical samples and monitor dynamic biochemical processes, achieving a high-throughput capability has remained a challenge. One approach to enhance throughput is the simultaneous detection of multiple samples (MacNamara et al., 1999; Kupče et al., 2021), drawing on the parallel imaging concept pioneered in MRI (Hyde et al., 1986; Pruessmann, 2006). Composite





detection via a bundle of isolated capillaries has enabled high throughput while reducing hardware complexity (Ross et al., 2001). Greater parallel independence relies on multiple radiofrequency (RF) coils for parallel excitation and reception, along with pulsed gradient fields (MacNamara et al., 1999; Hou et al., 1999) or alternative detection (Li et al., 1999) to separate the parallel detectors. Parallel NMR has been combined with classical pulse sequences (Wang et al., 2004), reaction kinetic measurements (Ciobanu et al., 2003), and dissolution dynamic nuclear polarization (Kim et al., 2016).

Integrating multiple coils into a single silicon chip has demonstrated portable NMR applications (Lei et al., 2020), employing time-interleaved pulses for decoupling. For full parallel and independent operation, each detector typically integrates individual RF coils, gradient coils, and shimming units (Cheng et al., 2022; Becker et al., 2023). However, practical limitations in electromagnetic shielding design, particularly in dense and highly composite arrays, result in field leakage and inter-channel coupling. RF decoupling schemes have been reported recently regarding both the excitation

and reception stages (He et al., 2024). The pulsed gradient field spillover among parallel detectors directly induces $B_0$ inhomogeneity, leading to spin dephasing, thus posing a challenge that remains to be addressed.

A straightforward approach to mitigate spin susceptibility to $B_0$ inhomogeneity is by transferring the spin state to longitudinal magnetization, e.g., $I_z$ for single spins or $I_z S_z$ for coupled spins. However, this proves challenging when the initial state is unknown, or time constraints exist. Spin locking via RF pulses is an effective strategy for controlling spins,

i.e., a long hard pulse with a defined phase is applied to preserve specific coherences, such as $I_x$ or $I_y$. In heteronuclear experiments, continuous wave spin-locking fields have been utilized to render proton multiple quantum coherence immune to $^1$H-$^1$H $J$ coupling (Grzesiek and Bax, 1995). Spin locking induced crossing has a wide range of applications, including preparation of singlet states (DeVience et al., 2013a, b; Rodin et al., 2018), excitation of long-lived states (Sonnefeld et al., 2022; Barskiy et al., 2017; DeVience et al., 2015), and low field spectroscopy (DeVience et al., 2021; Kovtunov et al.,

2014). Moreover, traditional spin-locking pulses have been adapted to address $B_0$ and $B_1$ inhomogeneity in quantifying the rotating frame relaxation time (Jiang and Chen, 2018; Gram et al., 2021).

In this study, we used optimal control to design cyclic pulses that preserve the desired coherence, such as $I^-$ for a single spin and $I^- S^+$ for a coupled spin pair. The coherence-locking by optimal control (CLOC) pulses exhibit robustness against a range of $B_0$ drifts, effectively mitigating field spillover effects in parallel NMR experiments. Multiple CLOC blocks

are inserted to safeguard specific coherence transfer pathways when the pulse sequence incorporates a series of gradient pulses. Using this protection idea, we demonstrated that the parallel HSQC experiment retains its SNR through coherence locking.



## 2 Results and Discussion

Three experimental scenarios were considered for the HSQC experiments to provide an initial view. Fig. 1a shows the
first case, which serves as a reference: a standard HSQC pulse sequence is used in which gradient pulses with a ratio of
$2 : 2 : -1$ are applied to select the $S^+ \rightarrow S^+ \rightarrow I^+ \rightarrow I^-$ coherence transfer pathway. Fig. 1b depicts the second scenario,
in which an additional set of gradient pulses $G_C$ was introduced. These gradients, independent of those in Fig. 1a, were
set to a ratio of $2 : 2 : 1$ and temporally shifted relative to the primary one to induce spillover effects. In Fig. 1c, the third
scenario incorporates three CLOC blocks along with $G_C$ into the HSQC sequence, the first and second CLOC blocks were
applied to $^{13}$C and the third was applied to $^1$H.

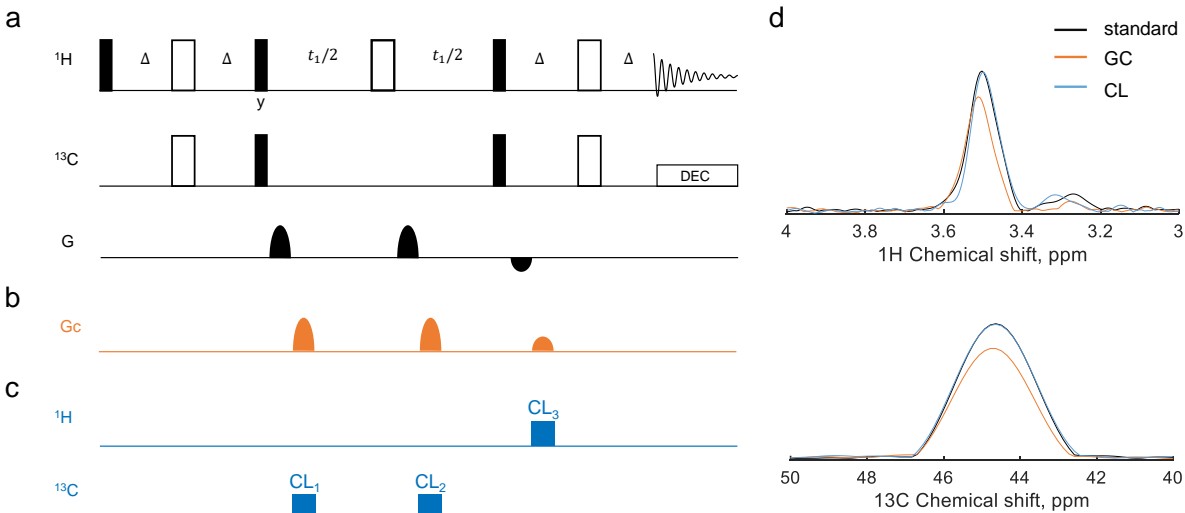

**Figure 1.** View of the three experimental HSQC scenarios. (a) The standard HSQC pulse sequence, in which the gradient pulse ratio
is set to $2 : 2 : -1$ to select the $S^+ \rightarrow S^+ \rightarrow I^+ \rightarrow I^-$ coherence pathway. (b) Additional gradient pulses from a second detector, set at a
ratio of $2 : 2 : 1$, are incorporated into (a) as coupled gradients. (c) CLOC pulses are introduced to (a), aligned with the coupled gradient
pulses. Specifically, $CL_1$ and $CL_2$ are applied to $^{13}$C, while $CL_3$ is applied to $^1$H. (d) The 1D projected spectra corresponding to the
three experimental scenarios, obtained using 0.6 M glycine in $D_2O$.

Experiments were conducted for all three scenarios using glycine as the test sample, and the resulting spectra are shown
in Fig. 1d. Compared to the reference spectrum from the standard HSQC experiment, the gradient coupling scenario
resulted in a 20% signal loss. In HSQC, a coherence pathway is refocused only if the net effect of all gradient pulses is



zero. Additional gradients can disrupt this balance, causing dephasing and signal loss, which can happen when separate

gradient-enhanced experiments are run in parallel at two detector sites. However, including CLOC blocks resulted in a signal intensity equal to the reference, indicating that the relevant coherences were preserved. Different gradient ratios are utilized in a parallel probe if multiple detectors execute different experiments or employ the same sequence but select distinct coherence pathways. The gradient spillover disrupts the intended coherence, and coherence lockingcan mitigate the dephasing effect in these scenarios. Note that if two detectors use identical sequences with the same gradient pulse

ratio, the total gradient ratio remains unchanged. Consequently, no gradient defense mechanisms are required, as shown in Fig. S2b.

Following the overview, we detail the gradient coupling issue and the coherence-locking compensation solution. Here, a two-detector array is shown in Fig. 2a, in which each detector is equipped with a Helmholtz-like gradient coil and a stripline RF coil. Due to the proximity of the two detectors, there is direct field spillover from one detector to the other.

The simulation results quantifying this coupling effect are given in Fig. S1, and the measured gradient spillover ratio is shown in Fig. 2c. Firstly, the maximum gradient strength of the parallel probe was measured using the pulsed gradient spin echo (PGSE) experiment, for a sample of 10% $H_2O$/90% $D_2O$ with a diffusion constant (Holz and Weingartner, 1991) $D = 1.9 \times 10^{-9}$ $m^2 \cdot s^{-1}$. The ratio of the signal to the maximum value is a function of the applied gradient amplitude (Stejskal and Tanner, 1965),

$$80 \quad \ln(I_g/I_0) = -[\gamma^2 \delta^2 G^2 (\Delta - \delta)]D \tag{1}$$

where $\delta = 1$ ms is the gradient pulse length, $\Delta = 7$ ms is the interval between two gradient pulses, and $\gamma = 2.675 \times 10^8$ $rad \cdot s^{-1} \cdot T^{-1}$ is the proton gyromagnetic ratio. Curve fitting yielded a maximum gradient of $G_{max} = 103.06$ G/cm, see Fig. 2b.





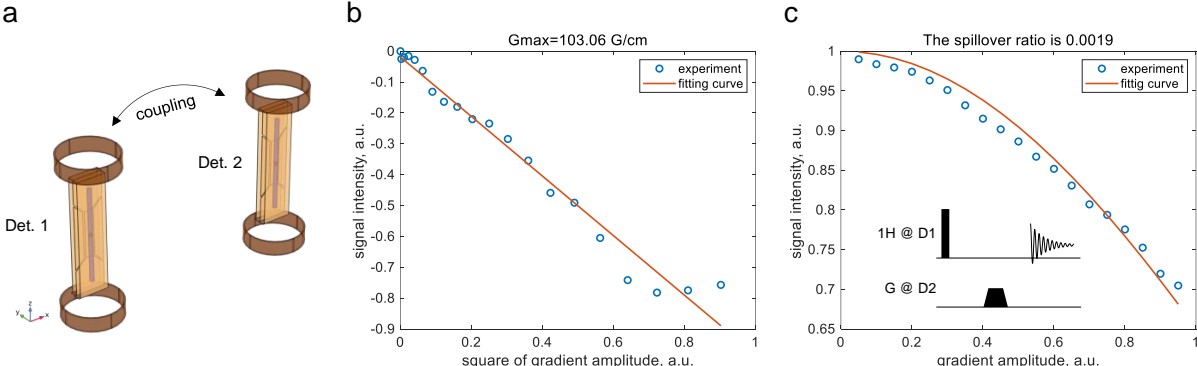

**Figure 2.** Measurement of the gradient spillover ratio, using 10% $H_2O$/90% $D_2O$. (a) Two-detector array. (b) Signal intensity as a function of gradient amplitude in a PGSE sequence, the maximum gradient amplitude was calculated to be $G_{max} = 103.06$ G/cm at a 100% gradient. (c) Signal intensity as a function of coupling gradient amplitude in a pulse-acquisition experiment, the gradient spillover ratio was calculated to be $R_G = 1.9 \times 10^{-3}$. The circles and the solid lines represent the experimental data and the fitted curves, respectively.

The gradient spillover ratio was then measured in a pulse-acquisition experiment, while a gradient pulse was applied

to the second detector, see the insert of Fig. 2c. The ratio of the signal to the maximum value is a function of the applied gradient amplitude,

$$\frac{I_g}{I_0} = \frac{\sin(kg)}{kg} \tag{2}$$

where $k = l/2 \cdot \gamma \cdot R_G \cdot G_{max} \cdot \int g(t)\, dt$, $l = 6.5$ mm is the detection zone length, and $\int g(t)\, dt = 0.9$ ms is the time integral of the trapezoidal gradient pulse. The gradient spillover ratio was determined by curve fitting as $R_G = 1.9 \times 10^{-3}$, see Fig. 2c.

Depending on the gradient pulses used, the impact of such a spillover can differ, for example the gradient coupling resulted in 20% signal loss in an HSQC experiment as shown in Fig. 1. Strong gradients are essential for diffusion experiments in which a coupled gradient of $0.20$ G/cm can be induced in the neighboring detector, making the coupling issue critical. Here, we propose a compensation scheme utilizing RF pulses to mitigate the gradient spillover. The idea is to protect desired coherences from unwanted field gradients using CLOC pulses, which are time-aligned with gradient pulses

from neighboring detectors. For example, a gradient pulse applied to detector 1 can be compensated by simultaneously applying a CLOC pulse to detector 2. This scheme was demonstrated with a PGSE experiment and a parallel HSQC experiment.





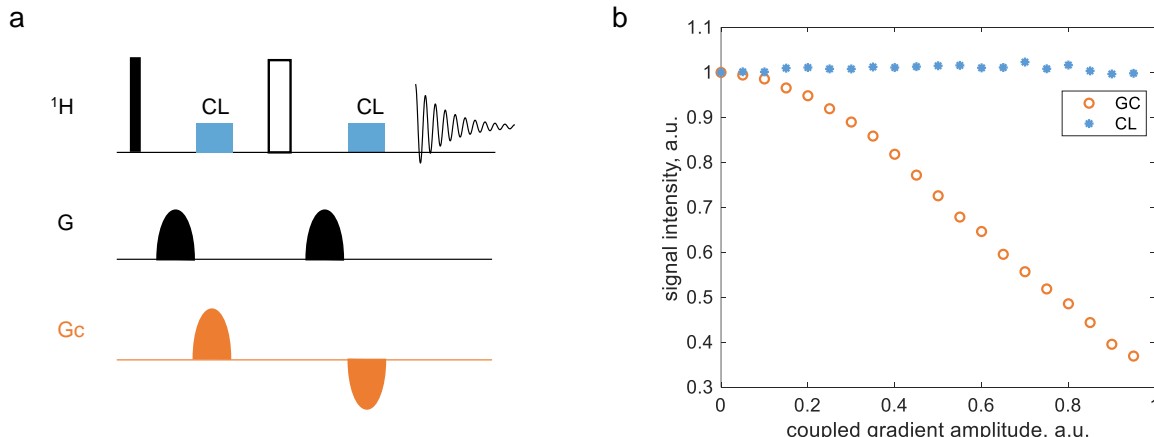

**Figure 3.** Coherence-lock test in PGSE using 0.6 M glycine in $D_2O$. (a) The pulse sequence, where the $^1H$ pulse and G were applied to detector 1, and the $G_c$ was applied to detector 2. (b) The signal intensity of the water peak, normalized to its value when $G_c = 0$, was plotted as a function of $G_c$, with G fixed at 10%.

As shown in Fig. 3a, an additional gradient, $G_C$, was introduced as the coupling component in a standard PGSE sequence, temporally shifted relative to the primary gradient. When $G_C$ disrupted the strength balance on either side of the inversion pulse, two CLOC pulses were applied, aligned with each block of $G_C$, to counteract gradient spillover. These CLOC pulses preserved spin coherence, protecting $I^+$ during the first $G_C$ period and $I^-$ during the second. Alternatively, the pulses can enable effective cyclic propagation (i.e., $\mathbf{U} = \mathbf{1}$), which is a stricter condition and was selected as our design target. Details of pulse optimization are provided later. The proposed coherence lockingin PGSE was tested using 0.6 M glycine in $D_2O$, with results shown in Fig. 3b. The $G_C$ amplitude was swept from 0 to 95% in the pulse program while the $G$ was fixed at 10%, and the water peak intensity was extracted and normalized to its value at $G_C = 0$ for comparison. Compared to the large signal loss in the gradient coupling case, the CLOC pulse effectively restored the signal.



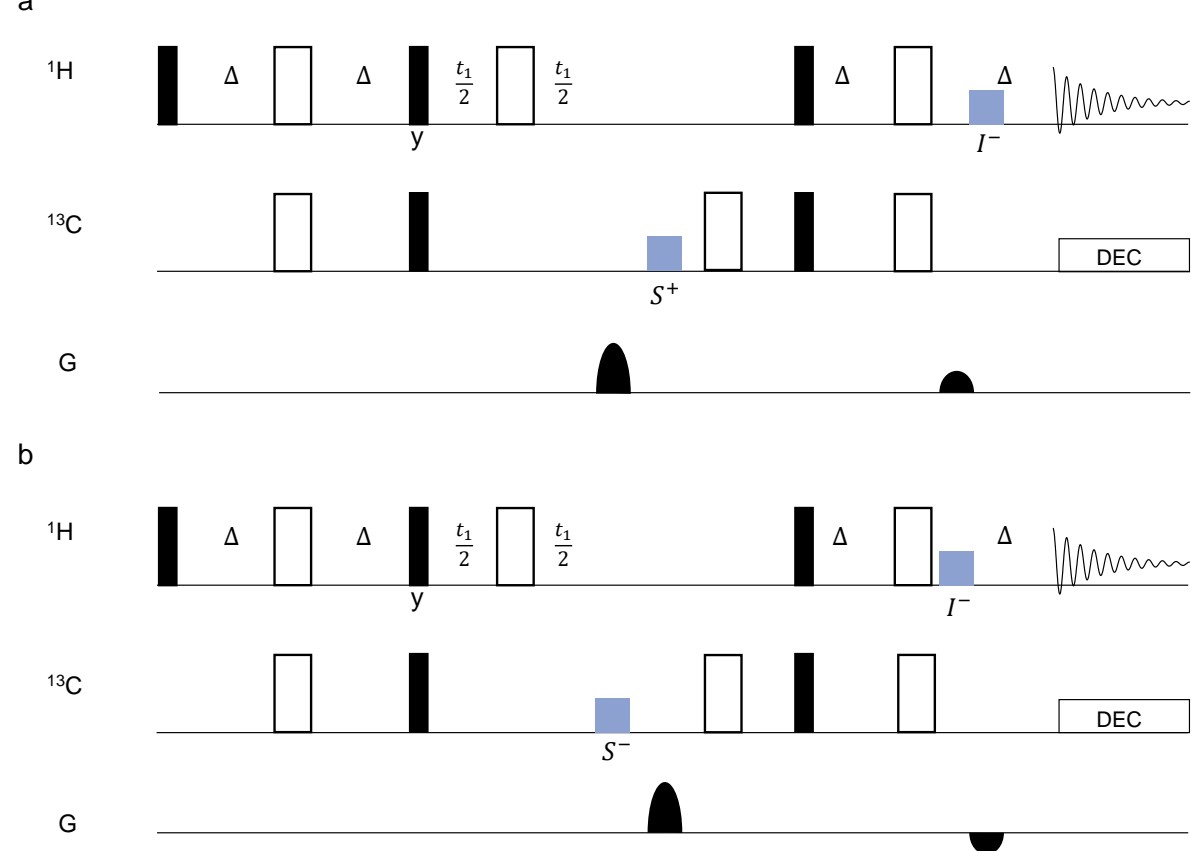

**Figure 4.** The scheme for gradient pulse compensation in a parallel HSQC pulse sequence involves blocks (colored in blue) indicating the CLOC pulse. The gradient pulse ratio in detector 1 (a) was set to 4:1 to select the $S^+ \rightarrow I^-$ pathway. In detector 2 (b), the gradient pulse ratio was set to 4:-1 to select the $S^- \rightarrow I^-$ pathway. The black blocks represent $\pi/2$ pulses and the white blocks represent $\pi$ pulses, with all phases set to 0 unless specifically noted. $\Delta = 1/4J$. The protected coherences are labeled below the CLOC blocks.

A pulse sequence tailored for parallel HSQC is illustrated in Fig. 4, in which the fundamental HSQC sequences were implemented in detectors 1 and 2. The gradient ratio in detector 1 was set to $4 : 1$ to select the $S^+ \rightarrow I^-$ pathway (Fig. 4a), while in detector 2, it was adjusted to $4 : -1$ to select the $S^- \rightarrow I^-$ pathway (Fig. 4b). The compensation CLOC pulse, indicated in blue, was applied in detector 1 while the gradient pulse was applied in detector 2, and vice versa. The gradient pulses in parallel detectors were executed with a slight time delay to allow the insertion of the CLOC pulses. Each compensation pulse was applied to lock onto the relevant coherence. For instance, the first CLOC block in detector



1 was applied on $^{13}$C to protect $S^+$ and its $J$ coupling-induced product $I_z S^+$. The protected coherences are labeled below the CLOC pulses in Fig. 4. With this approach, the CLOC pulse must be designed to compensate the gradient spillover (effectively a range of frequency offsets) for a desired coherence that is to be protected.

In the pulse optimization, both the source and target states were specified as the locked coherence. When a gradient pulse is applied, its shape and duration remain fixed, while its amplitude can vary. The optimization of CLOC pulses adheres to this principle. Specifically, the gradient pulse contributes to a time-dependent drift Hamiltonian $\mathbf{H}_g(t) = A \sin(\pi t/\tau) \mathbf{H}_{z0}$, where $\mathbf{H}_{z0}$ represents the Zeeman Hamiltonian under a 1 T field. The CLOC pulse should align with a gradient pulse with a fixed shape and duration. To cover a maximum $B_0$ drift of $\pm 0.06$ Gauss measured with the parallel probe, the maximum $B_0$ drift of $\pm 0.25$ Gauss was specified for the CLOC pulse. Therefore, multiple time-dependent drifts were included in the optimization to account for both spatial and temporal variations in $B_0$ drifts. The RF amplitude in the $^1$H channel was set to 6 kHz with a $\pm 20\%$ $B_1$ inhomogeneity, covering a 7 kHz bandwidth and simultaneously a maximum $B_0$ drift of $\pm 0.25$ Gauss, corresponding to $\pm 1.07$ kHz offset. For the $^{13}$C channel, the RF amplitude was adjusted to 4 kHz with a $\pm 15\%$ $B_1$ inhomogeneity, covering a 6 kHz bandwidth and simultaneously a maximum $B_0$ drift of $\pm 0.25$ Gauss. Pulse optimization can be targeted at a single spin if the $J$ coupling can be ignored or at coupled spins if the $J$-coupling is strong enough to be significant. Considering that concurrent optimization involves parameters from multiple nuclei, resulting in a large model, the strategy was to build optimal control for a single spin and test its decoupling effect within the designed bandwidth. The decoupling effect was quantified using average Hamiltonian theory (Waugh, 1982). The $J$-coupling scale factor is given by:

$$\chi = \text{norm}(\bar{\mathbf{c}}) \tag{3}$$

where $\mathbf{c}$ is the time-dependent $J$-coupling tensor in the toggling frame, defined by the RF pulse plus resonance offset (see Supplementary Note 4). Fig. 5 displays the $\chi$ values for CLOC pulses designed for universal locking of $I^+$, $I^-$, and $I_z$ of a single spin. A single CLOC pulse reduces $J$-coupling by less than 10% across the designed resonance offset and $B_1$ inhomogeneity range, see Fig. 5a and Fig. 5b. Given a pulse duration of 1 ms, heteronuclear $J$-couplings smaller than or on the order of hundreds of Hz can be ignored as $\chi J \tau \ll 1$. This treatment also applies to coupling with a third spin when studying $^1$H and $^{13}$C, such as $J_{CN}$ and $J_{HN}$ in proteins (LeMaster and Richards, 1985; Liu and Prestegard, 2009). Fig. 5c shows residual coupling exists when two CLOC pulses are applied simultaneously to $^1$H and $^{13}$C. However, it does not imply the decoupling fails as $\chi$ sums all the items of the J-coupling tensor. One can also calculate the coherence-locking efficiency when double-quantum coherences require protection. The coherence-locking for double-quantum coherence was tested by simulating an HMQC sequence, as detailed in Supplementary Note 7.



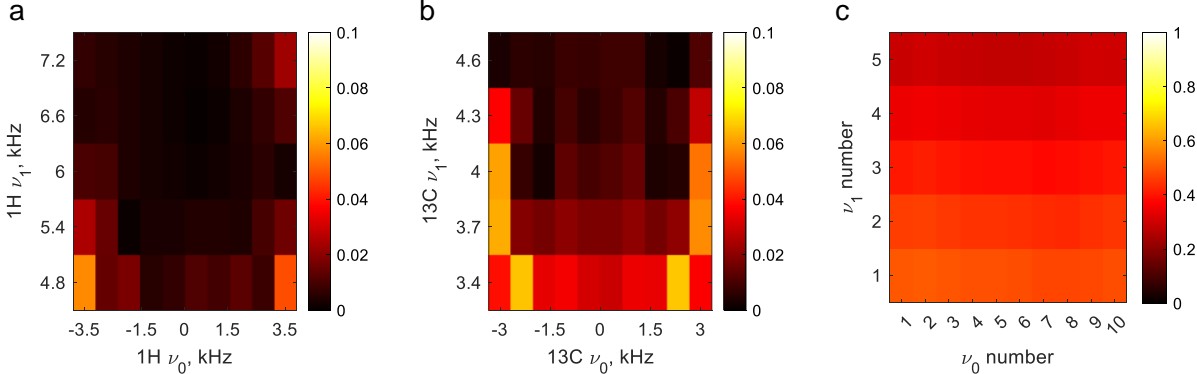

**Figure 5.** Scale factor $\chi$ of the $J_{HC}$ coupling as a function of resonance offset and $B_1$ amplitude. (a) The CLOC pulse is applied to $^1$H while $^{13}$C is on resonance. (b) The CLOC pulse is applied to $^{13}$C while $^1$H is on resonance. (c) The CLOC pulses are applied simultaneously to $^1$H and $^{13}$C, with $\nu_0$ on both channels aligned, and $\nu_1$ on both channels aligned.

Using this strategy, the simulated locking efficiency of the optimal control pulses is depicted in Fig. 6. The sample was segmented into 12 voxels along the z-direction, and the spin trajectory for each voxel was computed while simultaneously executing the coupled gradient, which induced a maximum $B_0$ drift of $\pm 0.25$ Gauss for the CLOC pulse. Fig. 6a illustrates that the ensemble spin states start from $I^-$ and oscillate between $-1$ and $1$ while remaining confined within a narrow range, and go back to $I^-$ at the end of 1 ms gradient pulse. Fig. 6b and Fig. 6c show the evolution of ensemble spin states starting from $S^+$ and $I^-S^+$, respectively. Additionally, Fig. 6d-f present the coherence evolution with the same initial states but without coherence looking for comparison, indicating coherence dephasing caused by the coupled gradient.



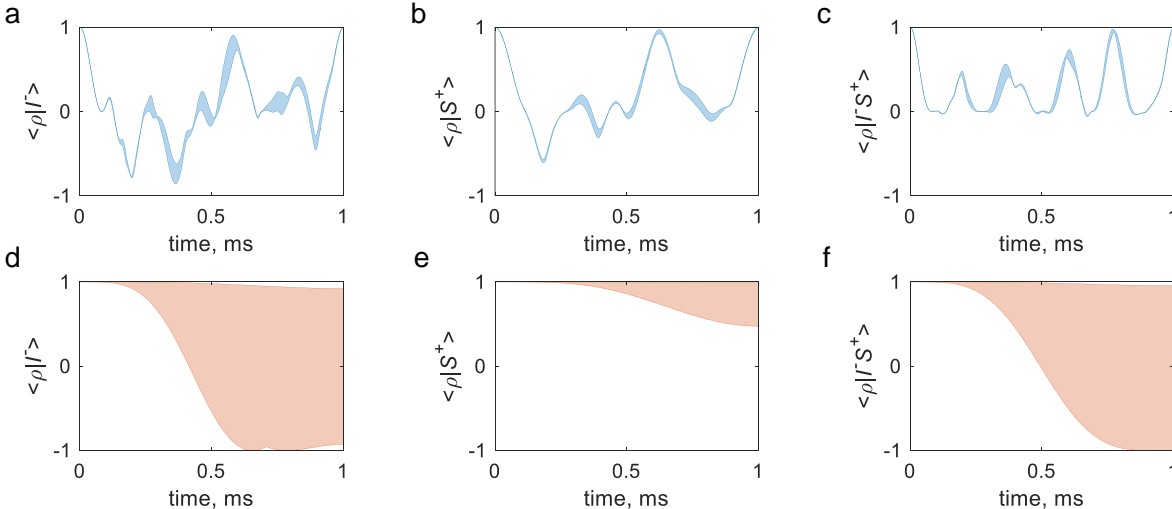

**Figure 6.** Ensemble spin trajectory considering the gradient field spillover dephasing. (a,b,c) represent the trajectories starting from $I^-$, $S^+$ and $I^-S^+$ with the CLOC pulses applied to $^1$H, $^{13}$C and $^1$H & $^{13}$C, respectively. (d,e,f) correspond to the respective trajectories without CLOC pulses. The real values of the inner product are displayed. The $^1$H and $^{13}$C were on resonance, and the $J$ coupling constant was 145 Hz. The RF amplitudes were 6 kHz for $^1$H and 4 kHz for $^{13}$C.

The parallel HSQC pulse sequence was tested using a parallel probe (see Methods for details), two detectors of the probe were used, with 0.6 M glycine in $D_2O$ in detector 1 and 0.3 M glucose in $D_2O$ in detector 2. The resulting spectra are presented in Fig. 7, where the 1D projections of the $^1$H and $^{13}$C signals from the 2D spectrum are displayed to illustrate the signal intensity for three cases. The single-detector scenario (grey) serves as a reference, while the orange lines represent amplitude-suppressed signals caused by gradient spillover-induced dephasing. In contrast, the blue lines correspond to the results of parallel operation with coherence-locking, demonstrating signal intensity recovery and effective dephasing compensation. Suppose the CLOC pulses effectively achieve broadband locking for both the $^1$H and $^{13}$C spin coherences, such that each peak in the coherence-locking case has the same intensity as in the reference case. However, in Fig. 7b, slight intensity differences are observed between the coherence-locking results and the reference, particularly for the peaks at 3.82 ppm and 3.89 ppm in the $^1$H dimension, correlating to 72.77 ppm and 79.07 ppm in the $^{13}$C dimension, respectively. Since the pulse optimization did not account for homonuclear coupling ($^1$H-$^1$H and $^{13}$C-$^{13}$C), coherence-locking efficiency is degraded, especially when the J-coupling constant is comparable to the offset difference between two peaks. A comparison of glucose HSQC spectra with and without homonuclear coupling is provided



in Fig. S7. The discrepancies also arise from the chemical shift dependence of coherence-locking efficiency. For instance, each CLOC pulse introduces slight phase distortion; if each pulse has a fidelity of 95%, the combined fidelity of two pulses reduces to 90%. The robustness of CLOC pulses encompasses factors including bandwidth, $B_1$ inhomogeneity, and

gradient amplitude, which are detailed in Supplementary Note 5. In addition, coherence-locking can also be influenced by RF coupling, when a CLOC pulse is applied in detector 1, weak RF signals (about 1% at 500 MHz (He et al., 2024)) may transfer from detector 1 to detector 2, distorting the spin state. To mitigate the RF coupling, the CLOC pulse was adjusted to relatively lower amplitudes, i.e., 6 kHz for $^1$H and 4 kHz for $^{13}$C. The corresponding power levels were 0.43 W for the $^1$H channel, and 2.3 W for the $^{13}$C channel on detector 2, making the coupling effect more significant in the $^{13}$C channel,

highlighting the challenges of coherence locking for low-sensitive nuclei when a large bandwidth is required.

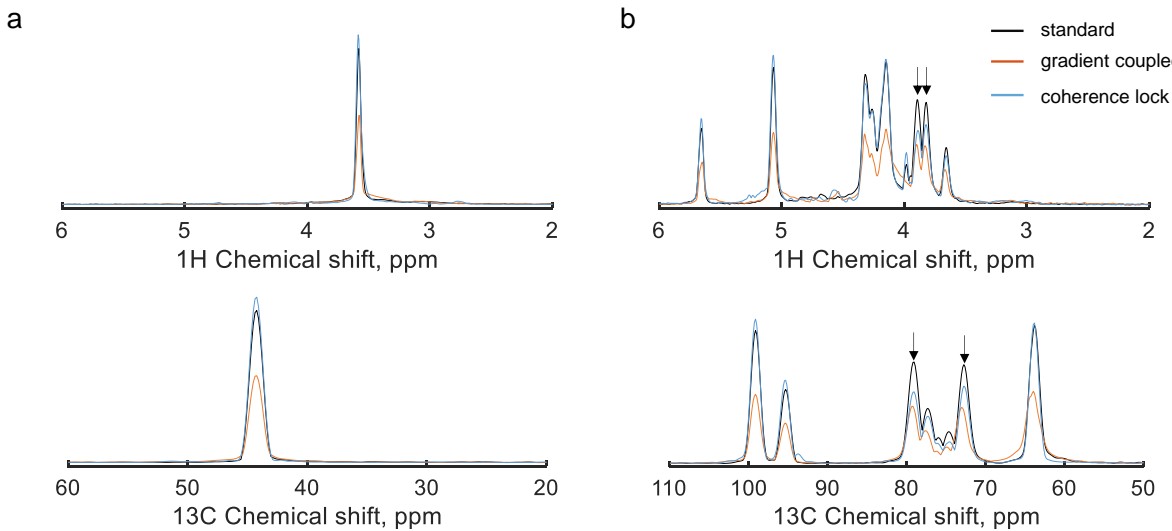

**Figure 7.** Experimental parallel HSQC spectra of glycine (a) and glucose (b) respectively, using the pulse sequence in Fig. 4. The grey lines represent the results without gradient coupling, as a reference. The orange lines display the spectrum under gradient spillover (no compensation), and the blue lines display the results with coherence-locking compensation.

## 3   Conclusion

We utilized CLOC pulses to safeguard specific coherences from being affected by gradient spillover in a parallel NMR scenario. This approach ensures effective compensation for the gradient-induced phase shifts, maintaining coherence



and signal integrity across parallel detectors. We evaluated the effectiveness of this approach through a parallel HSQC experiment.

Although we tested the decoupling effect of specific optimal pulses, general optimal pulses could potentially average out the $J$ coupling due to their "noise-like" waveforms. The overall strategy is to establish the locking of a single spin and validate its decoupling, thereby avoiding the complexity of a coupled spin model. While the CLOC pulses protect the targeted coherences, allowing coherence evolution to be neglected during this period, it should be excluded when calculating the J-coupling evolution delay. The limitation of extending this scheme to higher parallelism also lies in relaxation decay, which may become significant as more alternating gradient pulses join.

This work discussed conducting the same pulse sequence in two detectors with different gradient ratios. When the detectors perform different experiments, such as HMQC on detector 1 and HSQC on detector 2, the relaxation delay on each detector can be adjusted to synchronize the parallel pulses. However, because the acquisition is no longer simultaneous, addressing the coupling between the pulse and FID in the acquisition stage is beyond the scope of this work.

Due to electromagnetic and temperature fluctuations and spin-spin interactions, developing coherence protection methods for manipulating quantum qubits is essential (Miao et al., 2020). Examples include spin-locking of an electron spin for radiofrequency magnetometry (Loretz et al., 2013), suppression of intrabath interactions and inhomogeneity using sequenced pulses (Waeber et al., 2019), and extension of coherence time via continuous microwave driving (Ramsay et al., 2023). The optimal control-assisted coherence-locking demonstrated in this work offers an alternative approach for designing coherence protection protocols.

## 4 Methods

For sample preparation, a 10% $H_2O$/90% $D_2O$ solution was prepared to measure the gradient strength and gradient spillover ratio. Two solutions were prepared in $D_2O$ (99.9%): a 0.6 M labeled glycine solution and a 0.3 M labeled D-glucose solution for the parallel HSQC experiment. The same glycine solution was also used for the PGSE experiment. The samples were loaded into syringes and manually pumped into the individual fluidic chambers of the dedicated detector. The $D_2O$, [13]C-labeled glycine and [13]C-labeled D-glucose were purchased from Sigma-Aldrich.

Experimental validation was performed using a 4-detector parallel NMR probe (Voxalytic GmbH), as shown in Fig. S11, which was installed in a Bruker AVANCE NEO 11.7 T ([1]H frequency 500.13 MHz) NMR system (Bruker BioSpin GmbH). For demonstration, 2 of the 4 detectors were used, each double-resonant ([1]H/[13]C) and equipped with an independent



single-axis pulsed field gradient. The two gradient channels were powered by the Bruker GREAT micro-imaging amplifiers using a customized cable splitting the three amplifiers from a single cable to three individual cables (Bruker). Pulse calibration was conducted for each detector individually. In detector 1, a hard 90° $^1$H of pulse length 7.4 μs was applied at 20 W, a hard 90° $^{13}$C pulse of length 25 μs was applied at 25 W, with a corresponding value in detector 2 of 6.1 μs for $^1$H and 19.0 μs for $^{13}$C. The power levels were then scaled down for the CLOC pulses with low amplitudes. The HSQC contained 256 $t_1$ increments, each with 1 scan of 1024 data points. A total of 32 dummy scans were executed to stabilize the spin system before data collection, the relaxation delay is 1 second and the receiver gain is 10. The sweep width was 10 ppm for $^1$H and 150 ppm for $^{13}$C. Experiments explicitly run in parallel were done using the multireceive option in TopSpin 4.1.3. The parallel HSQC experiments were repeated twice for good shimming quality: data from detector 1 was collected with global shimming focused on detector 1, and data from detector 2 was collected with shimming focused on detector 2.

The magnetic field simulation was conducted with the finite element method software COMSOL MultiPhysics 6.1 (COMSOL AB, Sweden), and the simulated data was processed with Matlab 2023b (MathWorks Inc., USA). The pulse optimization and spin dynamics calculations were completed with Spinach v2.8 (Hogben et al., 2011), the detailed setting for optimal control is provided in Supplementary Note 3.

## Acknowledgements

M.H. is supported by the Joint Lab Virtual Materials Design (JLVMD) of the Helmholtz Association, Germany. J.G.K. acknowledges support from the ERC-SyG (HiSCORE, 951459). N.M., B.L., and J.G.K. acknowledge partial support from CRC 1527 HyPERiON. The authors acknowledge the support of the Helmholtz Society through the program Materials Systems Engineering. M.H. acknowledges support from the state of Baden-Württemberg through the bwHPC BwUniCluster 2.0. We acknowledge Dr. Thomas Oerther (Bruker BioSpin GmbH Co. KG) for support in running the GREAT imaging gradient amplifiers for parallel NMR spectroscopy experiments. We appreciate Prof. Ilya Kuprov's suggestions on Spinach simulations.



## Author contributions

225  J.G.K. and N.M. conceived the pulse compensation idea. M.H. performed the simulations with input from all coauthors. M.H. and N.M. drafted the manuscript. M.H., N.M. and D.B. conducted the experiments. All authors reviewed and refined the manuscript. J.G.K., B.L. and N.M. provided supervision and secured the funding.

## Competing interests

230  J.G.K. serves on the editorial board of *Magnetic Resonance* and is a shareholder of Voxalytic GmbH, a spinoff company that produces and markets microscale NMR devices. The other authors declare no competing interests.



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
