# Peer review of "Coherence locking in a parallel NMR probe defends against gradient field spillover"

_Magnetic Resonance, 2025_

## Author Response (AR2)

**Response to Reviewers**

March 21, 2025

**Reviewer 1**

This is a clever idea to use spin locking to essentially desensitize desired coherences from unwanted gradient spillover effects from a gradient set adjacent to the one which is being actively used. I have only a few minor suggestions to the manuscript.

We're grateful for the reviewer's positive feedback and for providing us with valuable comments.

1. Reference to parallel imaging in the introduction is not really appropriate. For example SENSE (Pruessman) uses measured coil sensitivity maps to reconstruct undersampled data from a single sample and multiple detectors, so this is a very different scenario from the one here.

Thank you for pointing out this difference between parallel NMR and multi-detector MRI, the relevant literature has been removed.

2. The authors often talk about spillover from one detector to another, by which they mean one coil/gradient combination to another, but i think this would be better phrased as gradient spillover, or the effects of gradient spillover, to separate from RF spillover.

We agree that gradient coupling and RF coupling jointly exist in the parallel probe, and it's necessary to distinguish the two effects. "Spillover" has been revised to "gradient spillover," and RF coupling was also clarified as needed throughout the article.

3. How generalizable is this approach to a much more complicated spin system, ie not just an IS heteronuclear one. Presumably spin locking would only be applicable to certain coherences at certain evolution times, or is this not correct. It would be good to see some discussion of this topic at the end of the paper.

Thank you for pointing this out.

Regarding coherence, the optimization of the coherence-locking pulse considered only a single spin, but the target is to generate a cyclic pulse ($U = \mathbb{I}$) that universally protects $I^+, I^-$ and $I_z$. Therefore, it is feasible to universally lock single-spin coherence ($I^+, I^-$) in

a more complex spin system. Universally locking double-quantum ($IS$) coherence is also theoretically feasible by applying two universal locking pulses simultaneously, as supported by the HMQC simulation results (Supplementary Note 7). However, locking higher-order heteronuclear coherence could be challenging in practice when multiple coherence-locking pulses are applied simultaneously. In this case, decoupling of the spin-spin coupling ($J$-coupling) must be examined, and RF coupling can become significant.

Regarding evolution time, the optimal control pulse duration is fixed at 1 ms, a representative gradient pulse duration in gradient-enhanced NMR. To adapt to a longer duration (a few ms) or shorter duration, the RF amplitude should be scaled down or up accordingly, along with the resulting covered bandwidth.

Finally, in the conclusion, we have added a statement (highlighted in blue) on the generalizability of coherence locking.

4. The authors state the pulse optimization does not account for homonuclear (HH or XX) coupling. Is this theoretically possible assuming a certain coupling constant, or does the problem become intractable for realistic spin systems?

Theoretically, it's feasible to compensate for the homonuclear coupling in the optimal control. The method is that two spins are included in the model, and a J-coupling term is specified in the Hamiltonian. The cost of compensation is extra RF power. However, it is not feasible to average out the homonuclear J-coupling between two spins when their chemical shift difference is comparable to the J-coupling constant. This can be understood using the average Hamiltonian theory. Suppose the two spins are labeled as $I_1$ and $I_2$, the Hamiltonian is given by

$$\mathbf{H} = \mathbf{H}_{\mathrm{z}} + \mathbf{H}_{\mathrm{J}} + \mathbf{H}_{\mathrm{rf}} \tag{1}$$

$$\mathbf{H}_{\mathrm{z}} = \omega_1 I_{1z} + \omega_2 I_{2z} \tag{2}$$

$$\mathbf{H}_{\mathrm{J}} = 2\pi J(I_{1x}I_{2x} + I_{1y}I_{2y} + I_{1z}I_{2z}) \tag{3}$$

$$\mathbf{H}_{\mathrm{rf}} = \omega_x(I_{1x} + I_{2x}) + \omega_y(I_{1y} + I_{2y}) \tag{4}$$

Considering that $[\mathbf{H}_{\mathrm{rf}}, \mathbf{H}_{\mathrm{J}}] = 0$, the RF pulse cannot solely average out the strong J coupling. When $|\omega_1 - \omega_2|$ is comparable to $J$, $[\mathbf{H}_{\mathrm{z}}, \mathbf{H}_{\mathrm{J}}]$ is close to 0, so the strong J

coupling cannot be averaged out by $\mathbf{H}_z$. As a result, the second-order spectrum cannot be fully recovered when the CLOC pulse is applied. When $|\omega_1 - \omega_2| \gg J$, it's possible to partially average out $\mathbf{H}_J$ by $\mathbf{H}_{rf} + \mathbf{H}_z$. The following figure shows the impact of homonuclear coupling on the coherence-locking efficiency of the $^{13}$C CLOC pulse. This figure has been included in the Supplementary Note 4.

[Figure]

Figure **R1.** Simulated coherence locking efficiency of the $^{13}$C CLOC pulse subject to a two-$^{13}$C spin system. The efficiency is defined as $\eta = \langle \rho_0 | \rho_T \rangle$, where $\rho_0$ is the initial state, $\rho_T$ is the final state. The $\delta\nu_0 = \omega_1 - \omega_2$, where $\omega_1$ is fixed at 0, and $\omega_2$ is swept from -3 kHz to 3 kHz, $J = 50$ Hz, and $\nu_1$ is the RF amplitude of the pulse. (a) $\rho_0 = I_{1x} + I_{2x}$, which commutes with $\mathbf{H}_J$. (b-c) $\rho_0 = I_{1x}$, using the CLOC pulse without J-coupling compensation (b) and with J-coupling compensation, where the RF amplitude is increased to 6 kHz (c).

5. The final paragraph of the conclusion is very strangely worded and could easily be removed.

Thank you for your suggestion, we have deleted these references.

**Reviewer 2**

Really nice work to mitigate gradient field spillover in parallel NMR spectroscopy, a technique aimed at increasing throughput for applications like drug discovery. The authors introduce "coherence-locking" (CLOC) pulses, designed via optimal control theory, to protect spin coherences from dephasing caused by gradient pulses in adjacent detectors. The study tests this compensation scheme in pulsed gradient spin echo (PGSE) and

heteronuclear single quantum coherence (HSQC) experiments, using a custom parallel NMR probe. A great idea and well executed. Although there are some limitations (related to spin-system specificity), the approach is innovative and potentially useful and therefore fully worthy of publication. The authors may wish to add a statement explaining the generalizability of approach.

We appreciate the reviewer's positive evaluation and kind suggestion regarding the generalizability of our work.

Regarding applying this approach to a general spin system, we also thank Reviewer 1 for raising similar points. While single-spin coherence can be universally locked in a general spin system, double-quantum ($IS$) coherences can be locked using two simultaneous locking pulses. However, locking higher-order coherences becomes challenging in practice, as multiple coherence-locking pulses can introduce significant RF coupling. Hence, the RF coupling may need to be jointly compensated alongside gradient spillover effects to extend to higher parallelism.

Another limitation is the impact of homonuclear coupling. Although optimal control can compensate for homonuclear coupling at the cost of extra RF power, if two spins have a very small chemical shift difference and the spectra are second order, coherence-locking pulses cannot compensate for this scenario.

In the conclusion, we have added a statement (highlighted in blue) for the generalizability of our approach.

**List of changes made in the manuscript**

1. Figure 2a has been relocated to Figure 1 to help readers follow the parallel experiment setup.

2. A statement acknowledging the limitations of our approach has been added to the conclusion to support a more general perspective.

3. The effect of homonuclear coupling is discussed in Supplementary Note 4 and referenced in the main text.

4. Some descriptions related to CLOC pulse optimization have been refined, and several typos have been corrected.